# Visual Representation Learning over Latent Domains

**Lucas Deecke, Timothy Hospedales & Hakan Bilen**
University of Edinburgh
{l.deecke,t.hospedales,h.bilen}@ed.ac.uk

## Abstract

A fundamental shortcoming of deep neural networks is their specialization to a single task and domain. While multi-domain learning enables the learning of compact models that span multiple visual domains, these rely on the presence of domain labels, in turn requiring laborious curation of datasets. This paper proposes a less explored, but highly realistic new setting called *latent domain learning*: learning over data from different domains, without access to domain annotations. Experiments show that this setting is challenging for standard models and existing multi-domain approaches, calling for new customized solutions: a sparse adaptation strategy is formulated which enhances performance by accounting for latent domains in data. Our method can be paired seamlessly with existing models, and benefits conceptually related tasks, e.g. empirical fairness problems and long-tailed recognition.

## 1 Introduction

Datasets have been a major driving force behind the rapid progress in computer vision research in the last two decades. They provide a testbed for developing new algorithms and comparing them to existing ones. However, datasets can also narrow down the focus of research into overspecialized solutions and impede developing a broader understanding of the world.

In recent years this narrow scope of datasets has been widely questioned (Torralba & Efros, 2011; Tommasi et al., 2017; Recht et al., 2019) and addressing some of these limitations has become a very active area of research. Two actively studied themes to investigate broader learning criteria are multi-domain learning (Nam & Han, 2016; Bulat et al., 2019; Schoenauer-Sebag et al., 2019) and domain adaptation (Ganin et al., 2016; Tzeng et al., 2017; Hoffman et al., 2018; Xu et al., 2018; Peng et al., 2019a; Sun et al., 2019b). While multi-domain techniques focus on learning a single model that can generalize over multiple domains, domain adaptation techniques aim to efficiently transfer the representations that are learned in one dataset to another.

Related themes have also been studied in domain generalization (Li et al., 2018; 2019b;a; Gulrajani & Lopez-Paz, 2020) and continual learning (Kirkpatrick et al., 2017; Lopez-Paz & Ranzato, 2017; Riemer et al., 2019), where the focus lies on learning representations that can generalize to unseen domains, and to preserve knowledge acquired from previously seen tasks, respectively.

While there exists no canonical definition for what exactly a visual domain is, previous works in multi-domain learning assume that different subsets of data exist, with some defining characteristic that allows them to be separated from each other. Each subset, indexed by $d = 1, \ldots, D$, is assigned to a pre-defined visual domain, and vice-versa multi-domain methods then use such domain associations to parameterize their representations and learn some $p_\theta(y|x, d)$.

In some cases domains are intuitive and their annotation straightforward. Consider a problem where images have little visual relationship, for example joint learning of Omniglot handwritten symbols (Lake et al., 2015) and CIFAR-10 objects (Krizhevsky & Hinton, 2009). In this case, it is safe to assume that encoding an explicit domain-specific identifier into $p_\theta$ is a good idea, and results in the multi-domain literature provide clear evidence that it is beneficial to do so (Rebuffi et al., 2018; Liu et al., 2019a; Guo et al., 2019a; Mancini et al., 2020).

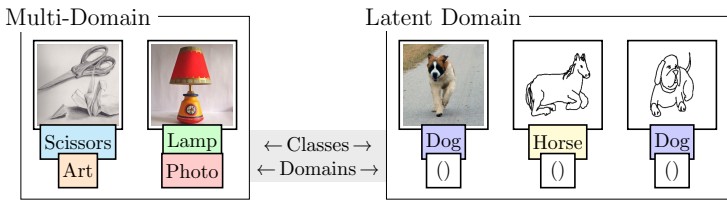

Figure 1: In multi-domain learning every sample has a domain label. Latent domain learning studies how models may best be learned without this information.

The assumption that domain labels are always available has been widely adopted in multi-domain learning; however this assumption is not without difficulty. For one, unless the process of domain annotation is automated due to combining existing datasets as in e.g. Rebuffi et al. (2017), their manual collection, curation, and domain labeling is very laborious.

And even if adequate resources exist, it is often difficult to decide the optimal criteria for the annotation of $d$: some datasets contain sketches, paintings and real world images (Li et al., 2017), others images captured during day or night (Sultani et al., 2018). Automatically collected datasets (Thomee et al., 2016; Sun et al., 2017) contain mixtures of low/high resolution images, taken with different cameras by amateurs/professionals. There is no obvious answer which of these should form their own distinct domain subset.

Moreover, the work of Bouchacourt et al. (2018) considers semantic groupings of data: they show that when dividing data by subcategories, such as size, shape, etc., and incorporating this information into the model, then this benefits performance. Should one therefore also encode the number of objects into domains, or their color, shape, and so on?

Given the relatively loose requirement that domains are supposed to be different while related in some sense (Pan & Yang, 2009), these examples hint at the difficulty of deciding whether domains are needed, and – if the answer to that is yes – what the optimal domain criteria are. And note that even if such assignments are made very carefully for some problem, nothing guarantees that they will transfer effectively to some other task.

This paper carefully investigates this ambiguity and studies two central questions:

1. Are domain labels always optimal for learning multi-domain representations?
2. How can models best be learned that generalize well over visually diverse domains, without domain labels?

To study this problem, we introduce a new setting (c.f. Fig. 1) in which models are learned over multiple domains without domain annotations — *latent domain learning* for short.

While latent domain learning is a highly practical research problem in the context of transfer learning, it poses multiple challenges that have not been previously investigated in connection with deep visual representation learning. In particular, we find that the removal of domain associations leads to performance losses for standard architectures due to imbalances in the underlying distribution and different difficulty levels of the associated domain-level tasks.

We carry out a rigorous quantitative analysis that includes concepts from multi-domain learning (Rebuffi et al., 2018; Chang et al., 2018), and find that their performance benefits do not directly extend to latent domain learning. To account for this lost performance, we formulate a novel method called *sparse latent adaptation* (Section 3.2) which enables internal feature representations to dynamically adapt to instances from multiple domains in data, without requiring annotations for this. Moreover, we show that latent domain methods appear to benefit single domain data and real world tasks, such as fairness problems (Appendix F), and long-tailed recognition (Appendix G).

## 2 LATENT DOMAIN LEARNING

This section provides an overview over latent domain learning and contrasts it against other types of related learning problems, in particular multi-domain learning.

Table 1: A comparison of latent domain learning versus unsupervised, latent source (Mancini et al., 2018) and semi-supervised DA, domain generalization, and multi-domain learning. $S_d$ denotes a labeled dataset from the $d$'th domain, $P_d$ are partially labeled, $U_d, \tilde{U}_d$ unlabeled.

| Setting | Domain Labels | Training Data | Evaluation Data |
|---------|---------------|---------------|-----------------|
| Unsupervised DA | Yes | $S_1, \ldots, S_D, U_{D+1}$ | $\tilde{U}_{D+1}$ |
| Latent Source DA | No | $S_{\text{mixture}}, U_{D+1}$ | $\tilde{U}_{D+1}$ |
| Semi-supervised DA | Yes | $S_1, \ldots, S_D, P_{D+1}$ | $U_{D+1}$ |
| Domain Generalization | Yes | $S_1, \ldots, S_D$ | $U_{D+1}$ |
| Multi-Domain Learning | Yes | $S_1, \ldots, S_D$ | $U_1, \ldots, U_D$ |
| Latent Domain Learning | No | $S_{\text{mixture}}$ | $U_1, \ldots, U_D$ |

## 2.1 PROBLEM SETTING

When learning on multiple domains, the common assumption is that data is sampled i.i.d. from a mixture of distributions $\mathbb{P}_d$ with domain indices $d = 1, \ldots, D$. Together, they constitute the data-generating distribution as $\mathbb{P} = \sum_d \pi_d \mathbb{P}_d$, where each domain is associated with a relative share $\pi_d = N_d/N$, with $N$ the total number of samples, and $N_d$ those belonging to the $d$'th domain. In multi-domain learning, domain labels are available for all samples (Nam & Han, 2016; Rebuffi et al., 2017; 2018; Bulat et al., 2019), such that the overall data available for learning consists of $\mathcal{D}_{\text{MD}} = \{(x_i, d_i, y_i)\}$ with $i = 1, \ldots, N$.

In latent domain learning the information associating each sample $x_i$ with a domain $d_i$ is not available. As such, domain-specific labels $y_i$ cannot be inferred from sample-domain pairs $(x_i, d_i)$ and one is instead forced to learn a single model $f_\theta$ over the latent domain dataset $\mathcal{D}_{\text{LD}} = \{(x_i, y_i)\}$. While latent domain learning can include mutually exclusive classes and disjoint label spaces $\mathcal{Y}_1 \cup \cdots \cup \mathcal{Y}_D$ (as in long-tailed recognition, see Appendix G), we mainly focus on the setting of shared label spaces, i.e. $\mathcal{Y}_d = \mathcal{Y}_{d'}$. For example a dataset may contain images of dogs or elephants that can appear as either photos, paintings, or sketches.

Latent domains have previously attracted interest in the context of domain adaptation, where the lack of annotations was recovered through hierarchical Hoffman et al. (2012) and kernel-based clustering (Gong et al., 2013), via exemplar SVMs (Xu et al., 2014), or by measuring mutual information (Xiong et al., 2014). More recent work corrects batch statistics of domain adaptation layers using Gaussian mixtures (Mancini et al., 2018), or studies the shift from some source domain to a target distribution that contains multiple latent domains (Peng et al., 2019b; Matsuura & Harada, 2020). Latent domain learning however differs fundamentally from these works: Table 1 contains a comparison to existing transfer learning settings.

A common baseline in multi-domain learning is to finetune $D$ models, one for each individual domain (Rebuffi et al., 2018; Liu et al., 2019a). This requires learning a large number of parameters and shares no parameters across domains, but can serve as a strong baseline to compare against. We show that in many cases, even when domains were carefully annotated, a dynamic latent domain approach can surpass the performance of such domain-supervised baselines (see Section 4).

## 2.2 OBSERVED VS. UNIFORM ACCURACY

Consider a problem in which the data is sampled i.i.d. from $\mathbb{P} = \pi_a \mathbb{P}_{d_a} + \pi_b \mathbb{P}_{d_b}$, i.e. two *hidden* domains. When domain labels are not available in the data, a standard strategy is to treat all samples equally, and measure the observed accuracy:

$$\text{OAcc}[f] = \mathbb{E}_{(x_i, y_i) \sim \mathbb{P}}[\mathbb{1}_{y_{f(x_i)} = y_i}], \tag{1}$$

where $y_f$ denotes the class assigned to sample $x_i$ by the model $f$, and $y_i$ its corresponding label for training. The OAcc has a problematic property: if $\mathbb{P}$ consists of two imbalanced domains such that $\pi_a \geq \pi_b$, then the performance on $d_a$ dominates it. For example if $d_a$ has a 90% overall share, and the model perfectly classifies this domain while obtaining 0% accuracy on $d_b$, then OAcc would still assume 0.9, hiding the underlying damage to domain $d_b$.

This motivates alternative formulations for latent domain learning, to anticipate (and account for) imbalanced domains in data. If it is possible to identify some semantic domain labeling (as typically included in multi-domain/domain adaptation benchmarks), one can compare performances across individual subgroups. This allows picking up on domain-specific performance losses which traditional metrics (such as OAcc) fail to capture.

Where this is possible, we therefore propose to also measure latent domain performance in terms of *uniform accuracy* which decouples accuracies from relative ground-truth domain sizes:

$$\text{UAcc}[f] = \frac{1}{D} \sum_{d=1}^{D} \mathbb{E}_{(x_i, y_i) \sim \mathbb{P}_d}[\mathbb{1}_{y_f(x_i) = y_i}]. \tag{2}$$

Returning to the above example, a uniform measurement reflects the model's lack of performance on $d_b$ as UAcc $= 0.5$. Once again note while ground-truth domain annotations are required in order to compute uniform accuracy, these are never used to train latent domain models.

## 3 METHODS

To enable robust learning in the new proposed setting, we formulate a novel module called *sparse latent adaptation* which can adaptively account for latent domains. Section 3.1 reviews adaptation strategies popular in the multi-domain context, which our method extends (and generalizes).

### 3.1 LATENT ADAPTATION

When domain labels $d$ are available (not the case in latent domain learning) one strategy established by Rebuffi et al. (2017) is to modulate networks by constraining the layerwise transformation of residual networks (He et al., 2016) $\Phi(x) = x + f(x)$ to allow at most a linear change $V_d$ per each domain from some pretrained mapping $\Phi_0$ (with $f_0$ in every layer), whereby $\Phi(x) - \Phi_0(x) = V_d x$. Note the slight abuse of notation here in letting $x$ denote a feature map with channels $\mathcal{C}$. Rearranging this yields:

$$\Phi(x, d) = x + f_0(x) + \sum_{d=1}^{D} g_d V_d(x), \tag{3}$$

with a domain-supervised switch that assigns corrections to domains, i.e. $g_d = 1$ for $d$ associated with $x$ and 0 otherwise. Each $V_d$ is parametrized through 1x1 convolutions, and $f_0$ denotes a shared 3x3 convolution obtained e.g. on ImageNet (Deng et al., 2009). This builds on the assumption that models with strong general-purpose representations require minimal changes to adapt to new tasks (Bilen & Vedaldi, 2017), making learning each $V_d$ sufficient, while $f_0$ remains as is. Such adaptation strategies have been successfully used in few shot learning (Li et al., 2021) and NLP (Stickland & Murray, 2019) to restrict the number of learnable parameters there.

In latent domain learning access to $d$ is removed, resulting in two new challenges: we have no a priori information about the right number of corrections $\{V_d\}$, and we cannot use $d$ to decide which one of these to apply.

To mitigate the lack of domain labels $d$, first we assume that input data is constituted by $K$ latent distributions $\mathbb{P}_k$. Second we propose to replace the switch $g_d$ with a learnable gating mechanism $g_1(x), \ldots, g_K(x)$ that assigns each sample $x$ to latent domains as follows:

$$\Phi(x) = x + f_0(x) + \sum_{k=1}^{K} g_k(x) V_k(x), \tag{4}$$

The gates $g_k$ control which convolution is applied to which sample $x$, and correspond to a categorical variable over $K$ categories, i.e. $0 \leq g_k \leq 1$ and $\sum_k g_k = 1$. Note in particular how parametric dependency of $\Phi$ on $d$ is removed. How to best choose $K$ is discussed in more detail in Section 4.

While we motivate our latent domain module from learning over multiple domains, the main goal is not to recover the domain labels annotated in some datasets. When optimizing some loss (standard cross-entropy in the classification case), there is no guarantee that the learned $V_k$ will correspond to an annotated visual domain and many additional factors (shape, pose, color, etc.) can enter them as

well. Latent domain models are simply optimized to produce the lowest training error, and in fact seldom recover ground-truth domains (c.f. Fig. 5). Note the broader concept presented here may in principle also be incorporated with other multi-task concepts (Perez et al., 2018; Guo et al., 2019a), adaptation strategies however stand out due to their methodological simplicity.

Different options exist for parametrizing the gating function $g\colon \mathcal{X} \to \mathcal{G} \subseteq \mathbb{R}^K$. An ideal gating mechanism for latent domain learning would fulfill two seemingly incompatible properties: be able to filter latent domains in some layers (requiring a discrete gate), but also share parameters between related latent domains in other layers (smooth gates). The next section proposes how this can be resolved without requiring task relationships (Vandenhende et al., 2020) or outer optimization loops (Wu et al., 2018) through the use of sparseness.

## 3.2 SPARSE LATENT ADAPTERS (SLA)

We parameterize the gating function $g$ with a small linear transformation $W\colon \mathcal{C} \to \mathbb{R}^K$ that constitutes the pre-activation $q = W\varphi(x)$ within the gates, where $\varphi\colon \mathcal{X} \to \mathcal{C}$ denotes an average pooling projection onto the channels.

A crucial choice is whether the activation for $q \in \mathbb{R}^K$ should map to a *discrete* space $\mathcal{G} = \{0,1\}^K$ or a *continuous* $\mathcal{G} = [0,1]^K$ in which the $V_k$ are shared.

We propose a different strategy that lets gates be smooth when appropriate, but a threshold $\tau$ allows for sparse (or discrete) outputs $f_\tau(q) = [q - \tau(q)]_+$ with $[\cdot]_+ = \max(0, \cdot)$. Crucially $f_\tau$ can be solved in a differentiable manner (Martins & Astudillo, 2016) by sorting $q_1 \geq \cdots \geq q_K$, solving $k^* = \max\{k \mid 1 + kq_k > \sum_{j \leq k} q_j\}$ and computing $\tau = [(\sum_{j \leq k^*} q_j) - 1]/k^*$.

Consider $q = [0.1, 1.0, 0.5]$ for which sparse activation results in $f_\tau(q) = [0.0, 0.75, 0.25]$ while softmax yields $[0.202, 0.497, 0.301]$. Sparse activation filters out $q_1$, while sharing between $q_2$ and $q_3$. We may now define:

$$\mathrm{SLA}(x) \triangleq x + f_0(x) + \sum_{k=1}^{K} \left[ f_\tau \circ W \circ \varphi(x) \right]_k V_k(x), \tag{5}$$

where $[\cdot]_k$ picks the $k$'th element of the gating sequence. To the best of our knowledge sparse activation strategies were never previously employed for expert models in computer vision and have so far been restricted to the NLP setting (Deng et al., 2017; Peters et al., 2019). Note SLA generalizes residual adaption (Rebuffi et al., 2017; 2018), which is recovered by setting $K = 1$.

While gating is subject to complex interactions such as negative transfer (Rosenbaum et al., 2019), our ablations in Table 5 clearly show that taking a sparse perspective – which allows the model to assume either continuous or discrete forms – outperforms the alternative of a priori fixing either smoothness through self-attention (Lin et al., 2017b), or discrete Gumbel-based sampling (Jang et al., 2016). Note this choice between discrete (Veit & Belongie, 2018; Guo et al., 2019b) and continuous mechanisms (Shazeer et al., 2017; Sun et al., 2019a; Wang et al., 2019) delineates previous work that employs differentiable gates.

A softmax-activated model can in principle also learn to suppress individual preactivation components by letting some $q_k$ go to $-\infty$. This however requires either learning extra calibration parameters at every layer, defining a hard cutoff value (Shazeer et al., 2017) (thereby removing differentiability), or very large row-norms within the linear mapping $W$—a highly unlikely outcome given the several mechanisms found in state-of-the-art models (in particular weight decay, norm-penalties, or BN (Ioffe & Szegedy, 2015)) which act as direct counterforces to this.

## 4 EXPERIMENTS

We evaluate our proposed methods on three latent domain benchmarks: Office-Home, PACS, and DomainNet (c.f. Fig. 6, which shows example images from these benchmarks). The main goal here is not to compare to existing multi-domain or domain adaptation methods that these datasets were initially designed for, but to study our two central research questions: whether domain labels are useful for effectively learning over multiple domains, and whether one can learn such representations without domain labels.

Table 2: Per-domain performance on Office-Home. Multi-domain (MD) baselines use domain annotations, and latent domain (LD) models do not. Best overall performance underlined; best latent domain performance in bold.

| | Type | Params | A | C | P | R | OAcc | UAcc |
|---|---|---|---|---|---|---|---|---|
| Proportion $\pi_d$ | — | — | *15.57* | *28.01* | *28.48* | *27.95* | — | — |
| RA (Rebuffi et al., 2018) | MD | 8.8 mil | 48.05 | 76.12 | 80.74 | 67.78 | 70.73 | 68.17 |
| Domain-Adv. (Ganin et al., 2016) | MD | 7.8 mil | 55.14 | 72.85 | 81.98 | 68.81 | 71.57 | 69.70 |
| 4×ResNet26 | MD | 24.8 mil | 52.47 | 79.95 | 85.02 | 70.01 | 74.34 | 71.86 |
| ResNet26 | LD | 6.2 mil | 50.10 | 76.80 | 78.83 | 63.36 | 69.47 | 67.27 |
| ResNet56 | LD | 14.0 mil | 52.26 | 78.47 | 80.80 | 66.34 | 71.66 | 69.47 |
| RA (Rebuffi et al., 2018) | LD | 6.9 mil | 58.44 | **79.15** | 81.55 | 72.13 | 74.65 | 72.82 |
| MLFN (Chang et al., 2018) | LD | 7.6 mil | 50.72 | 78.81 | 81.36 | 64.56 | 71.18 | 68.86 |
| MMLD (Matsuura & Harada, 2020) | LD | 7.8 mil | 59.63 | 67.89 | 81.16 | 74.35 | 72.19 | 70.76 |
| SLA | LD | 7.6 mil | **60.72** | 78.05 | **83.73** | **77.08** | **76.70** | **74.89** |

We also examine a recent fairness benchmark (see Appendix F), and show that SLA improves robustness under single domain long-tailed distributions (Appendix G). All experiments were implemented in PyTorch (Paszke et al., 2017).[1]

**Optimization** In all experiments, we couple our method with a ResNet26 model pretrained on a downsized version of ImageNet that was used in previous work by Rebuffi et al. (2018). In SLA only gates and corrections are learned, the residual backbone $f_0$ remains fixed at its initial parameters, which implicitly regularizes the model (Rebuffi et al., 2017). Training is carried out for 120 epochs using stochastic gradient descent (momentum parameter of 0.9), batch size of 128, weight decay of $10^{-4}$, and an initial learning rate of 0.1 (reduced by 1/10 at epochs 80, 100).

All experiments follow the preprocessing of Rebuffi et al. (2017; 2018), alongside standard augmentations such as normalization, random cropping, etc. Accuracies are averaged over five seeds.

Increasing the number of corrections $K$ within SLA results in small, consistent performance gains. As $K = 2$ already represents a solid boost from the baseline of having no adapters, we focus on this result in the main part, and report results for higher $K$ alongside variances in Appendix C.

**Office-Home** The underlying data contains a variety of objects classes (alarm clock, backpack, etc.) among four domains: *art*, *clipart*, *product*, and *real world* (Venkateswara et al., 2017). In Table 2 we show results for $d$-supervised multi-domain (MD) approaches: RA (Rebuffi et al., 2018), domain-adversarial learning (Ganin et al., 2016) and a baseline of 4×ResNet26, one for each domain. For latent domain (LD) baselines, we then learn a single ResNet26, this time as a latent domain model over all domains. Next, we couple SLA with the very same ResNet26.

Learning a single ResNet26 over latent domains with no access to $d$-labels significantly harms performance. This problem is not addressed by simply increasing the depth of the network: while accuracy improves slightly, a ResNet56 exhibits the same performance losses — in particular on the latent domains *product* (P) and *real world* (R).

While residual adaptation (RA) (Rebuffi et al., 2018) was shown to work extremely well in many multi-domain scenarios, performance here is sub-par, regardless of whether it accesses $d$ (MD: one $V_d$ per-domain) or not (LD). This likely results from linear modules being reserved for each $d$ when using annotations, enabling no native cross-domain sharing of parameters. When $d$ is hidden on the other hand, the model is forced to share a single linear adaptation module $V$ between all four hidden domains, without the flexible gating we propose in SLA.

Learning annotations through latent domain clustering and coupling this with domain-adversarial gradient reversal as in MMLD (Matsuura & Harada, 2020) increases performance relative to its $d$-annotated counterpart (Ganin et al., 2016). The increase is modest however, likely because enforcing domain-invariance on the gradient level negatively impacts the model's ability to discriminate between classes (Wang et al., 2020). Another related baseline is MLFN (Chang et al., 2018) which builds on ResNeXt (Xie et al., 2017) to define a latent-factor architecture that accounts for multi-

---

[1]Code is available at `github.com/VICO-UoE/LatentDomainLearning`.

Table 3: Results on PACS. Best performance underlined, best latent domain performance bold.

| | Type | Params | • A | • C | • P | • S | OAcc | UAcc |
|---|---|---|---|---|---|---|---|---|
| Proportion $\pi_d$ | — | — | *0.205* | *0.235* | *0.167* | *0.393* | — | — |
| RA (Rebuffi et al., 2018) | MD | 8.8 mil | 85.14 | 92.05 | 94.50 | 94.30 | 91.93 | 91.50 |
| Domain-Adv. (Ganin et al., 2016) | MD | 7.8 mil | 86.47 | 93.25 | 94.17 | 91.30 | 91.25 | 91.30 |
| 4×ResNet26 | MD | 24.8 mil | 88.41 | 95.53 | 94.34 | 95.71 | 93.94 | 93.50 |
| k-means+RA | LD | 8.8 mil | 83.84 | 92.10 | 94.75 | 93.01 | 91.21 | 90.93 |
| ResNet26 | LD | 6.2 mil | 85.27 | 94.55 | 93.85 | 94.98 | 92.70 | 92.16 |
| ResNet56 | LD | 14.0 mil | 86.96 | 94.34 | 95.15 | 95.34 | 93.36 | 92.95 |
| RA (Rebuffi et al., 2018) | LD | 6.9 mil | 89.86 | 93.90 | 95.56 | 93.91 | 93.35 | 93.31 |
| MLFN (Chang et al., 2018) | LD | 7.6 mil | 78.38 | 91.29 | 88.19 | 92.95 | 88.78 | 87.70 |
| MMLD (Matsuura & Harada, 2020) | LD | 7.8 mil | **89.93** | 92.26 | **96.25** | 94.34 | 93.20 | 93.27 |
| SLA | LD | 7.6 mil | 89.27 | **95.85** | 95.26 | 95.49 | **94.26** | **93.97** |

Table 4: Results on DomainNet, best performances in bold.

| | clipart | infograph | painting | quickdraw | real | sketch | OAcc | UAcc |
|---|---|---|---|---|---|---|---|---|
| $\pi_d$ | *0.082* | *0.088* | *0.123* | *0.294* | *0.295* | *0.118* | — | — |
| ResNet26 | 66.46 | 27.99 | 51.70 | 67.46 | 66.58 | 56.95 | 60.47 | 56.19 |
| ResNet56 | 69.08 | 29.55 | 53.85 | 68.61 | 68.60 | 58.42 | 62.19 | 58.01 |
| RA (Rebuffi et al., 2018) | 68.23 | 29.34 | 56.29 | 66.51 | 71.81 | 57.75 | 62.65 | 58.32 |
| MLFN (Chang et al., 2018) | 68.20 | 24.64 | 50.39 | **69.85** | 65.78 | 56.27 | 60.54 | 55.85 |
| SLA | **69.46** | **30.14** | **57.36** | 67.97 | **72.89** | **58.83** | **63.83** | **59.44** |

modality in data. Crucially where our method is fine-grained and shares convolutions at every layer, MLFN instead enables and disables entire network blocks, allowing us to outperform it.

SLA outperforms the currently available latent domain models by a consistent margin, and increases UAcc by 12.79% relative to ResNet26. Best performance is obtained when $K = D$, with performance being reducing slightly from overfitting of larger domains for $K > D$ (see Appendix C).

**PACS** The second experiment examines performance on the PACS dataset (Li et al., 2017). Crucially PACS domains (*art, cartoon, photo, sketch*) differ more markedly from one another (c.f. examples in Fig. 6), hence constituting an interesting latent domain problem.

Even for more distinct domains as in PACS, results in Table 3 show that SLA improves over existing baselines. The largest gains occur on smaller domains (e.g. *art*), where standard models suppress underrepresented parts of the distribution (see additional discussion on imbalanced distributions in Appendix G). Our method again surpasses the accuracy of 4×ResNet26, while requiring a fraction of the total parameters ($\sim 9.7$ mil for $K = 5$ vs. $\sim 24.8$ mil). The performance of SLA again continues to increase with larger $K$ (see Appendix C).

The performance increase from using a latent domain-adversarial approach (Matsuura & Harada, 2020) versus using domain-annotations (Ganin et al., 2016) confirms that learning domains alongside the rest of the network can be a better strategy than trusting in annotations. Our approach again improves over this, without requiring a clustering stage as in MMLD.

Results for k-means (using $D = 4$ centers and clustered on the feature level) and subsequent finetuning show that a two-stage strategy is suboptimal. This is not surprising since, similar to $d$-supervision via $g_d$ in $\Phi$ of eq. (3), clustering learns fixed switches that get used across all layers. In contrast to this in SLA we flexibly share or separate features individually at every layer (c.f. qualitative results in Fig. 3), synergizing only where appropriate.

**DomainNet** We also evaluate models on a large-scale benchmark called DomainNet (Peng et al., 2019a). This dataset contains 518 447 images from six domains (clipart, painting, photos, sketch, infographics, and quickdraw), with a total of $|\mathcal{Y}| = 345$ object classes. The optimization settings remain unchanged from those in previous sections.

Results are shown in Table 4. MLFN performs best on quickdraw, a domain that differs visibly from others (c.f. Fig. 6 for examples from each domain), and having entire network blocks dedicated to it seems to benefit performance. On all remaining domains, SLA outperforms existing models,

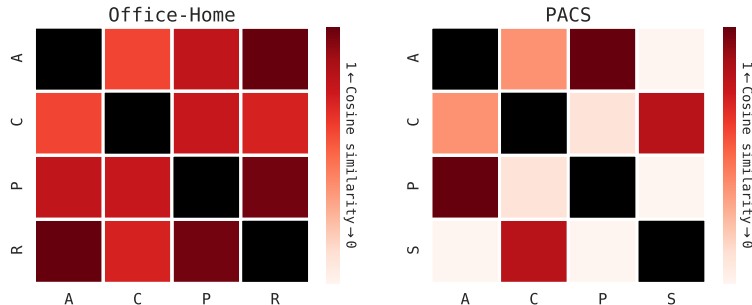

Figure 2: Cosine similarity between SLA gates for Office-Home (left) and PACS (right). PACS domains are more dissimilar, but similarities exist, e.g. (A)rt and (P)hoto.

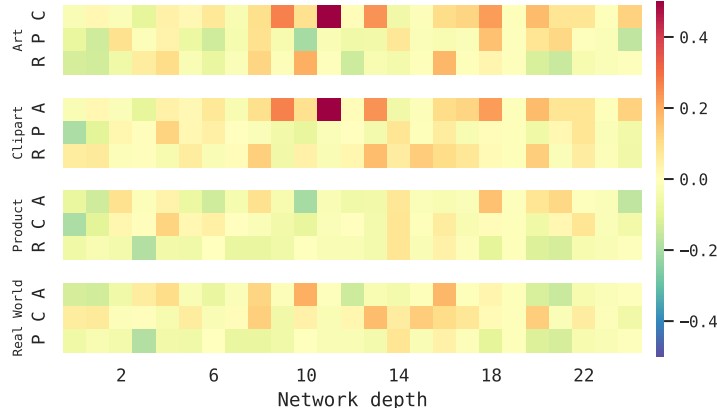

Figure 3: Layerwise correlation between SLA convolutions $V_k$ on Office-Home. Most correlations occur in the mid-to-late stages of the model.

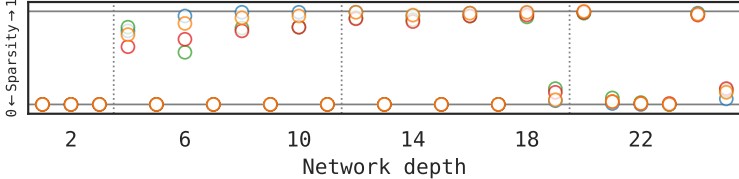

Figure 4: SLA sparsity (Office-Home); dotted lines show residual pooling transitions.

regardless of whether they were designed specifically for multi-domain problems, such as RA, or whether they are much deeper/parameter-intensive (ResNet56).

**Qualitative analysis**   We (i) compare global statistics of Office-Home and PACS domains as well as (ii) their per-layer treatment within SLA; (iii) analyze sparse gating, (iv) representations learned by SLA, and show that (v) our module shares between geometric properties (shape, pose, etc.).

i) Fig. 2: average cosine similarities of per-domain gating vectors $g \in \mathcal{G}^L$ across $l = 1, \ldots, L$ layers of ResNet26 show that Office-Home domains differ less than those in PACS.

ii) Fig. 3: layerwise measurements of $\mathrm{Corr}[g_l(x), g_l(x')]$ for $x, x'$ drawn from differing $d \neq d'$ for Office-Home. If inter-domain correlation is high, then similar corrections $V_k$ are responsible for processing samples from two domains. Across top layers of the network there is little correlation, presumably as low-level information associated with each domain is processed independently. In the mid to bottom stages correlation increases: these layers are typically associated with higher-order features (Yosinski et al., 2014; Mahendran & Vedaldi, 2016; Asano et al., 2020), and since label spaces are shared between latent domains, similar object-level features are required to classify objects into their respective categories.

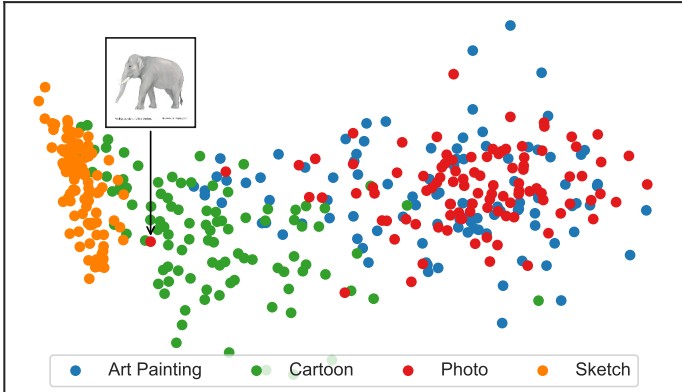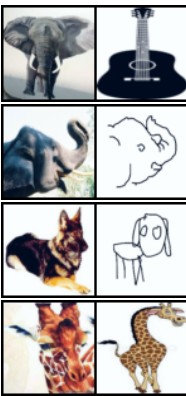

Figure 5: **Left**: PCA of samples represented by their SLA activation paths, colored by their ground-truth domain label as assigned in PACS. SLA shares parameters between visually similar domains *art* and *photo* (•,•), while isolating *sketch* (•). The arrow highlights one sample that has been labeled a *photo* in PACS. SLA categorizes it as a *cartoon* instead, a more adequate assignment for this particular image. **Right**: sample pairs from different domains ($d_i \neq d_j$) with matching SLA activations. Note their similar geometric properties.

iii) Fig. 4: sparse gates have the flexibility to either output singular activations (i.e. become fully discrete) or all-zero values (a continuous gate). We measure the per-layer sparsity $\mathbb{E}_{x \sim \mathbb{P}_d}[K - \|g_l(x)\|_0]/(K-1)$ where $\| \cdot \|_0$ counts values different from zero, finding sparsity of SLA to vary across model depth. Interestingly after each downsampling operation SLA tends to be relatively sparse, followed by a dense gate, then again a sparse one, and so forth. The model thus utilizes the extra flexibility resulting from sparse gates.

Due to PACS domains being relatively distinctive, the dataset is an interesting candidate for additional analysis in (iv) and (v) of how sparse adaptation handles the different ground-truth domains.

iv) Fig. 5 (left): gate vectors $g \in \mathcal{G}^L$ for samples from all four domains in PACS visualized by their principal components. SLA exhibits an intuitive clustering of human-annotated PACS domains: visually similar *art* and *photo* (•,•) cluster together. The manifold describing *sketches* (•) is arguably more primitive than those of the other domains, and indeed only maps to a small region. *Cartoon* (•) lies somewhere between sketches and real world images. This matches intuition: a cartoon is, more or less, just a colored sketch.

Fig. 5 also highlights one sample that shows an elephant that SLA places among the *cartoon* (•) domain, but which has been assigned a ground-truth domain label of *photo* (•) in the PACS dataset. The ground-truth label seems to have been annotated in error, but different from approaches that use $d$-supervision, our SLA processes latent domains on-the-fly and is therefore not irritated by this.

v) Fig. 5 (right): pairs of samples with similar gates. This shows that latent domains are indicative of more than ground-truth domain labels and extend to geometric similarities: pose, color, etc. of the samples are visibly related. Compare in particular the poses of elephants/dogs (second/third row).

## 5   CONCLUSION

In this paper we explored two questions: (i) whether domain associations are required for learning effective models over multiple visual domains and (ii) how multi-domain models may best be learned without depending on manually curated domain labels.

As has been shown, the performance of existing models *does* degrade without domain labels, raising doubts about their suitability for realistic problems that involve diverse data sources. As a remedy, we proposed a novel adaptation strategy which reclaims (and often exceeds) lost accuracy on latent domains, benefiting several problems where some notion of a domain (but no annotation) exists.

ACKNOWLEDGEMENT

HB is supported by the EPSRC programme grant Visual AI EP/T028572/1. TH was supported by EPSRC grant EP/R026173/1.

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

# APPENDIX

## A  DATASETS

Fig. 6 shows examples from the latent domain benchmarks evaluated in Section 4. The selected images have equivalent classes $y_d = y_{d'} \in \mathcal{Y}$ (for example chair for Office-Home), but different domains (e.g. $d = \{\text{art}, \text{clipart}, \text{product}, \text{real world}\}$).

These examples show that data from different domains often contain very different visual characteristics (compare e.g. photo vs. sketch for PACS), even when the object is the same. At the same time, other domains are more alike (e.g. art and photo), indicating that different amounts of sharing between per-domain parameters are required, which in SLA is facilitated by its gating mechanism.

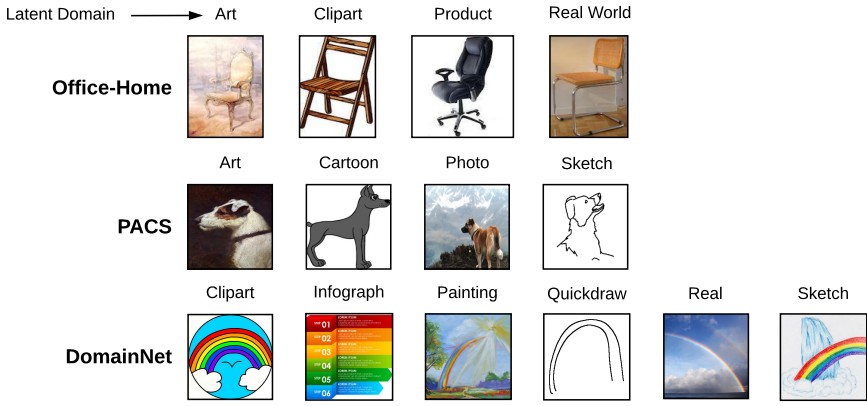

Figure 6: Datasets used for latent domain benchmarks: Office-Home, PACS, and DomainNet.

## B  RELATED WORK

Multi-domain learning relates most closely to our work. The state-of-the-art methods introduce small convolutional corrections in residual networks to account for individual domains (Rebuffi et al., 2017; 2018), which was recently extended to obtain efficient multi-task models for related language tasks Stickland & Murray (2019). Other work makes use of task-specific attention mechanisms (Liu et al., 2019a), attempts to scale task-specific losses (Kendall et al., 2018), or addresses tasks at the level of gradients (Chen et al., 2017). Crucially, these approaches all rely firmly on domain labels.

Our work is loosely related to learning universal representations (Bilen & Vedaldi, 2017), which was used as a guiding principle in designing more transferable models (Tamaazousti et al., 2019). However, these works also assume the presence of domain labels. Multimodal learning does not make this assumption, and was shown to benefit from accounting for latent semantic factors to match images (Chang et al., 2018), or from normalizing data in separate groups (Deecke et al., 2019). As we show in our experiments (see Section 4), latent domain learning however benefits from more customized solutions than these.

The proposed module gives rise to a differentiable dynamic network architecture, studied e.g. for reinforcement learning (Zoph & Le, 2017; Pham et al., 2018), Bayesian optimization (Kandasamy et al., 2018), or when adapting to new tasks (Mallya et al., 2018; Rosenfeld & Tsotsos, 2018). For such architectures, two components are commonly used: discrete Gumbel-based sampling (Jang et al., 2016), e.g. leveraged in dynamic computer vision architectures (Veit & Belongie, 2018; Sun et al., 2019a), or continuous self-attentive approaches (Lin et al., 2017b), which have been used successfully to scale expert models (Jacobs et al., 1991; Jordan & Jacobs, 1994) to large problem spaces (Shazeer et al., 2017; Wang et al., 2019).

From the perspective of algorithmic fairness, a desirable model property is to ensure consistent predictive equality across different identifiable subgroups in data (Zemel et al., 2013; Hardt et al., 2016; Fish et al., 2016). This relates to one of the goals in latent domain learning: to limit implicit model bias towards large domains, and improve robustness on small domains. Recent work explores connections between models and empirical fairness for visual recognition (Bagdasaryan et al., 2019; Hooker et al., 2020; Wang et al., 2020), different from our experiments however (see Appendix F) they focus their analysis on a setting in which annotations for protected attributes are available.

## C  VARIATION OF RESULTS

Fig. 7 displays variances of accuracies recorded over ten random initializations on Office-Home (left) and PACS (right). We generally found SLA to be robust to different optimization settings, and as a result observed variances are relatively low across experiments.

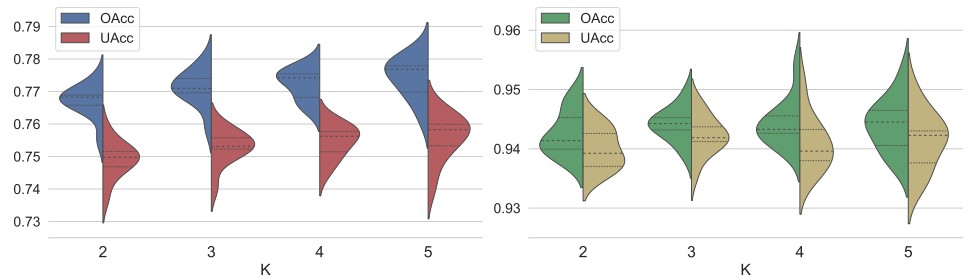

Figure 7: Variation of accuracies on Office-Home (left) and PACS (right).

Larger $K$ brings an improvement of around 0.5-1% in performance at the expense of a linear increase in learnable parameters (c.f. next section). While accuracy is improved by setting $K > 2$, gains appear to saturate in line with previous observations around network width (Xie et al., 2017).

## D MEMORY REQUIREMENTS

In SLA every layer contains $\mathcal{O}(K|\mathcal{C}| + K|\mathcal{C}|^2)$ parameters to parametrize gates and corrections $V_k$, respectively. This is however an extremely modest requirement, in particular because $f_0$ stays fixed: while a ResNet26 contains $\sim 6.2$ mil learnable parameters, even when setting $K = 5$ within SLA it has just 3.5 mil free parameters, and is a fraction of the number of parameters needed to parametrize four ResNet26 (around 24.8 mil parameters).

Note also that the complexity of solving sparse gates in SLA scales as $\mathcal{O}(K \log K)$, a negligible increase given the small $K$ required in our method.

## E ABLATION

Table 5: An ablation study for SLA. UAcc is shown (on Office-Home) with activations other than SparseMax (Martins & Astudillo, 2016) used in this paper. For all variants we fix $K = 2$.

| Gating Mechanism | UAcc |
|---|---|
| SLA | 74.89 |
| Smooth (Lin et al., 2017b) | 74.62 |
| Discrete (Jang et al., 2016) | 74.31 |

Replacing sparse gating within SLA registers a drop in performance, regardless of whether smooth or discrete mechanisms are used. Accuracies for soft and straight-through Gumbel-softmax sampling (Jang et al., 2016) were on par; we report straight-through sampling here.

We also ran experiments where we did not fix the residual backbone $f_0$ but updated its parameters alongside the learning of SLA. In line with what Rebuffi et al. (2017) report, this lead to overfitting and performance dropped to UAcc $= 73.53$.

## F FAIRNESS

Recent work elevated the role of small subgroups in data and examined model fairness on CelebA (Bagdasaryan et al., 2019; Wang et al., 2020; Hooker et al., 2020). Because such subgroups may be interpreted as constituting an individual latent domain component $\mathbb{P}_d$, they are an interesting candidate to evaluate our purpose-built SLA on.

The benchmark contains different labeled attributes (e.g. "brown hair", "glasses"), and is modified from the original dataset by hiding gender labels. Models are evaluated on all 39 remaining

Table 6: Average precision and bias amplification of SLA on the CelebA fair attribute recognition benchmark (Wang et al., 2020).

|  | ResNet18 | + SLA | ResNet34 | + SLA | ResNet50 | + SLA |
|---|---|---|---|---|---|---|
| mAP ($\uparrow$) | 71.76 | 73.22 *(+1.46)* | 71.33 | 73.98 *(+2.65)* | 74.52 | **75.03** *(+0.51)* |
| BA ($\downarrow$) | 0.025 | 0.014 | 0.022 | 0.009 | 0.012 | **0.008** |

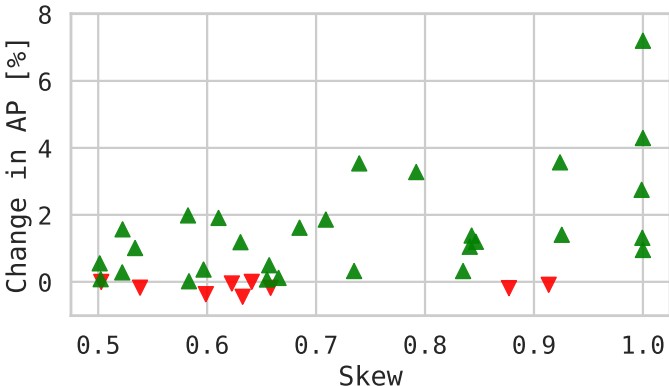

Figure 8: Change in AP between ResNet18 and ResNet18-SLA for different gender skews in CelebA attributes.

attributes, which subsequently experience varying amounts of gender skew. Framed as a latent domain problem we have $d = \{\text{female}, \text{male}\}$, but models have no access to this information.

The images used are the entire Aligned&Cropped subset (Liu et al., 2015) over which we finetune residual models, replacing only the fully-connected layer of the network. We use the optimization settings introduced in Section 4 for 70 epochs with reductions at epochs 30, 40, and 50, selecting the best model on the validation split. This experimental setup is identical to previous work on empirical fairness (Wang et al., 2020; Ramaswamy et al., 2020), which however – different from our work – focused on learning models that have access to the gender-attribute $d$.

We evaluate per-attribute accuracy using mean average precision (mAP) and report bias amplification (BA) (Zhao et al., 2017). This compares the propensity of a model to make positive predictions (i.e. $f$ exceeds some threshold $t_+ \in [0,1]$) in the gender $g_y^*$ that appears most frequent within attribute $y$, compared to the true counted ratio of positive examples $y_+$:

$$\text{BA}[f] = \mathbb{E}_{x \sim \mathbb{P}_x}\left[\frac{\mathbb{1}_{f(x)>t_+|g_y^*}}{\mathbb{1}_{f(x)>t_+}}\right] - \mathbb{E}_{y \sim \mathbb{P}_y}\left[\frac{\mathbb{1}_{y=y_+|g_y^*}}{\mathbb{1}_{y=y_+}}\right], \tag{6}$$

where $t_+$ is optimized for on the validation split. For example if 60% of male examples are wearing glasses but under the model this is raised to a total of 65%, then bias is amplified by $\text{BA} = 0.05$.

We report performance for ResNet18, ResNet34, and ResNet50 in Table 6 and compare this to the same model with SLA inserted. SLA consistently raises both mAP and reduces bias, indicating that it relies less on spurious correlations in data to formulate its predictions.

In Fig. 8 we compare per-attribute skew toward either female or male (whichever is more frequent) to the gain in performance from ResNet18 to the same model but with SLA inserted. We observe a clear trend here, whereby SLA is able to raise performance the most in those attributes that experience the largest amounts of skew.

## G    LONG-TAILED RECOGNITION

Standard models often experience difficulty when some classes are heavily underrepresented. This problem has recently been studied in long-tailed recognition (Liu et al., 2019b; Cao et al., 2019) with

Table 7: Top-1 validation accuracy on imbalanced CIFAR benchmarks (Buda et al., 2018). SLA consistently improves performance for standard ERM as well as existing long-tail approaches.

| | | **Imbalanced CIFAR-10** | | | | |
|---|---|---|---|---|---|---|
| $\rho$ | ERM | ERM-SLA | Focal | Focal-SLA | LDAM | LDAM-SLA |
| 10 | 86.09 | **93.05** *(+6.96)* | 86.61 | 92.14 *(+5.53)* | 91.08 | 92.49 *(+1.41)* |
| 100 | 68.02 | **81.60** *(+13.58)* | 67.44 | 78.82 *(+11.38)* | 75.67 | 80.96 *(+5.29)* |
| | | **Imbalanced CIFAR-100** | | | | |
| $\rho$ | ERM | ERM-SLA | Focal | Focal-SLA | LDAM | LDAM-SLA |
| 10 | 65.16 | **70.76** *(+5.60)* | 64.56 | 70.60 *(+6.04)* | 66.08 | 69.61 *(+3.53)* |
| 100 | 45.44 | 48.46 *(+3.02)* | 45.19 | 48.39 *(+3.20)* | 51.25 | **55.06** *(+3.81)* |

benchmarks that modify CIFAR-10 and CIFAR-100 to an imbalanced version by dropping some classes (e.g. 6-10 for CIFAR-10) (Buda et al., 2018). The severity of the imbalance is described via the ratio $\rho = n_{\max}/n_{\min}$ between the largest and smallest classes.

Long-tailed distributions may be viewed as containing an underrepresented latent component with $\pi = 1/(1 + \rho)$, and previous results (c.f. Section 4) that fortified small latent domains within $\mathbb{P}$ motivate us to evaluate the imbalance setting more closely here.

Since our strategy is architecture-based, it can be combined with the most recent state-of-the-art (loss-based) techniques for long-tailed recognition: a label-distribution-aware margin loss with deferred reweighting (Cao et al., 2019), or reducing contributions from well-classified examples as in focal losses (Lin et al., 2017a). As Table 7 shows, adaptation via sparse gates acts as a regularizer on the underlying ResNet26, and consistently improves performance on long-tail benchmarks.

