# OpenReview forum: "Visual Representation Learning over Latent Domains"
_ICLR.cc/2022/Conference — ICLR 2022 Poster_

### Official Review · Reviewer_Yiib · 2021-10-28

**Correctness:** 2
**Technical Novelty And Significance:** 2
**Empirical Novelty And Significance:** 3
**Recommendation:** 3
**Confidence:** 2

**Main Review:**

[Strengths]

1. A new and interesting problem setting with practical values:
The new task introduced in this paper, i.e., latent domain learning, looks like a tweak of existing problems such as multi-domain learning at first glance, but is interesting and has great potential as it could be practically useful when learning and testing models with data from heterogeneous domains (e.g., web images crawled by predefined search keywords). I also agree that the definition of visual domains is often vague and contrived, thus believe the motivations of latent domain learning make sense.

2. Strong performance:
The proposed method outperforms multi-domain learning methods in the latent domain learning setting (although such a result is to be expected, the gap seems not large enough, and some scores of previous work look abnormal).


[Weaknesses]

1. Weak quality of writing:
The manuscript is overall readable but sometimes hard to follow, probably due to its weird terminology and expressions. Also,
- U^tilde in Table 1 is not defined.
- It is unclear why Eq. (2) is called “uniform” accuracy.
- The description for Eq. (3) is overly complicated.
- The operation denoted by stars is not defined appropriately, one may guess it stands for correlation though.

2. Limited novelty:
The proposed model LSA looks like an extension of the residual adapter (Rebuffi et al., 2018), dubbed RA in the paper. Both RA and LSA adopt and modify the well-known residual connection by adding domain-specific feature transforms. The main difference between them is that to choose appropriate feature transforms RA utilizes domain labels explicitly while LSA estimates the latent domain label of input through gating variables. The use of gating variables could be counted as a contribution, but they are not a new idea as they have been widely used in other fields of machine learning.
Also, I would note that some objections against RA in the paper look invalid and accordingly the contribution of this paper should be undervalued. As far as I understand the residual adaptation method (Rebuffi et al., 2018) aims at designing models that maximize parameter sharing across different domains; only a small number of parameters of their residual adaptation modules are learned in a domain-specific manner while the majority of their parameters are shared across domains.

3. Experiments:
Compared to RA, the improvement by the proposed method does not look sufficiently large. Also, the records of RA are weird: I wonder why its score is degraded when using domain labels, and how it is trained without domain labels in the latent domain learning setting.


**Summary Of The Paper:**

This paper introduces a new task called latent domain learning and proposes a baseline that extends an existing multi-domain learning method. The latent domain learning assumes that training data are sampled from different domains yet their domain labels are latent, and aims at learning models that generalize well to the domains. The proposed baseline deploys multiple parallel feature transform layers that are chosen on the fly through gating variables, with the hope that the gating variables learn to predict the latent domain label of input and choose feature transforms of the domain accordingly. This model demonstrates superior performance to existing multi-domain learning methods in the latent domain learning setting.

I recognize that latent domain learning is an interesting problem with great potential, and has many applications such as learning using web-crawled images. However, the proposed method looks limited in terms of novelty and the quality of writing is below the standard.


**Summary Of The Review:**

The new problem setting, latent domain learning, looks interesting and practically useful. However, the baseline model is limited in terms of novelty, its performance is not impressive (decent though), and the paper misleads readers about the important previous work (i.e., RA).
The paper has both pros and cons, but I would value novelty and correctness more, thus leaning towards rejection.

---

> ### Author Response · Authors · 2021-11-20
> **Re: Official Review of Paper228 by Reviewer Yiib**
>
> Thank you for your review. We felt there were a couple of misconceptions which we address below:
>
> * *“Weak quality of writing: The manuscript is overall readable but sometimes hard to follow”*
>
> The main terms used in this paper are widely accepted and we strictly follow standard terminology of e.g. Rebuffi et al. (2017/18), and Pan & Yang (2009). A lot of work went into writing this manuscript, and please note this was also positively highlighted by some of the other reviewers. Notation ($\tilde U$, etc.) has been updated in our revision.
>
> * *“It is unclear why Eq. (2) is called “uniform” accuracy”*
>
> Please see page 4 (second paragraph), where it states UAcc decouples domain sizes and their impact on the accuracy, such that each domain contributes to the reported metric equivalently.
>
> * *“The description for Eq. (3) is overly complicated”*
>
> This expression extends a standard formulation of Rebuffi et al. (2017/18), and motivates our conceptual extension to latent domains alongside abstractions needed for it. The notation of Eq. (3) and Eq. (4) was updated in our revision for a streamlined expression.
>
> * *“Gating variables could be counted as a contribution, but they are not a new idea”*
>
> Our manuscript 1) identifies a novel learning problem and 2) formulates a novel sparsely gated and dynamic model for this. Note that while sparse gating has been applied in NLP tasks, to the best of our knowledge ours is the first work to show it also benefits visual representation learning.
>
> * *”The paper misleads readers about the important previous work (i.e., RA)”*
>
> We strongly rebut this, as we make connections to RA very transparent, discuss RA at multiple points, and evaluate it extensively in our manuscript. Please note our criticisms extends to all MD methods, which cannot be applied in many realistic settings, and are therefore better handled by LD methods such as SLA.
>
> * *“Some objections against RA in the paper look invalid [..] only a small number of parameters of their residual adaptation modules are learned in a domain-specific manner while the majority of their parameters are shared across domains”*
>
> Let us clarify that SLA is highly parameter-efficient, as it freezes the residual backbone for transfer between object-level tasks (ablations for this are in Appendix C). We discuss the efficiency of SLA in detail on page 7 (paragraph 5) and in Appendix D. To emphasize this important point, we added parameters required in each model to Table 2/3.
>
> * *“Compared to RA, the improvement by the proposed method does not look sufficiently large.”*
>
> The performance improvements are clear. They range up to a substantial +7.71% (Table 2), and they are persistent across benchmarks (note an analysis of the variation of results has been added in Appendix E).
>
> * *“The records of RA are weird: I wonder why its score is degraded when using domain labels, and how it is trained without domain labels in the latent domain learning setting.”*
>
> When using domain labels (MD condition) RA trains one adapter per domain. Without domain labels (LD condition) RA adds a single adapter shared across all the domains. The reason RA+LD can outperform RA+MD is discussed on page 6 (second to last paragraph). Learning one RA for each domain may fit the train data better, but cannot improve generalization by exploiting parameter sharing. Learning one RA shared across all domains enforces parameter sharing, which is helpful to the extent that some domains are similar, but detrimental to the extent that there are differences between some. The ability to dynamically choose the amount of sharing is the key feature of SLA. And the fact that both RA+LD and RA+MD make too strong assumptions about the data distribution, and thus under perform, is a central criticism in our manuscript.
>
> * *"The baseline model is limited in terms of novelty”*
>
> Let us again highlight the main novelty of this work:
> 1. The introduction of latent domain learning as a new setting that has been overlooked in recent literature & is useful for practical problems (e.g. web-crawled data, fairness, imbalanced learning, c.f. Appendix F/G).
> 2. The formulation of SLA which generalizes RA and outperforms existing approaches across benchmarks. This is a first step towards the development of new latent domain methods, and we are hopeful it will lead to many new methods being proposed in future research.
> 3. Strong evidence that sparse gating can be very useful not just for NLP but also visual detection problems, in particular when it comes to parametrizing dynamic networks.
> 4. A detailed and extensive analysis of how SLA works and contributes to latent domain learning, e.g. analyzing which layers synergize with one another (Figure 3), or which examples are clustered apart/together in our dynamic model (Figure 5).

---

### Official Review · Reviewer_LPa6 · 2021-10-31

**Correctness:** 3
**Technical Novelty And Significance:** 2
**Empirical Novelty And Significance:** 3
**Recommendation:** 6
**Confidence:** 2

**Main Review:**

**Strengths**

1. The paper is well-written, easy to follow and understand.
2. The paper claims to propose a new setting for learning from multiple domains, where domain labels are unknown. This is an important problem since labeling domains can be challenging and/or expensive in real-world applications. While I am not extremely familiar with the field, according to the paper, prior works assume the knowledge of domain associated with a data point.
3. The idea of using sparse gating mechanism to allow for multiple transformations makes sense since it is possible that transformations learned across various domains are useful for predicting image category.
4. Qualitative and quantitative experiments seem to corroborate that authors' proposed method is learning a sensible domain representation.

**Weaknesses**

1. The experimental setup used in the paper, as well as the baselines are not properly justified. I specifically don't understand
    - Why is the ResNet backbone kept frozen, and not fine-tuned along with rest of the parameters?
    - Why do the authors use same parameters across different methods/models? A fair way to compare across methods would be to perform cross-validation across range of hyper-parameters and report the performance on the test set.
2. It is a bit surprising to me that all the proposed multi-domain approaches match perform worse when compared to latent-domain? I'd imagine that a model having a domain information for each datapoint would set an upper bound on the performance when compared to methods with no domain information. Is the presence of domain information hurting for the model performance?
3. Although the paper states that the reported numbers are averaged over five random initializations, the standard errors are not reported
4. I would like to hear authors' take on connection between self-supervised learning approaches, and given latent domain learning. The SSL methods learn image representations using self-similarity. In an extreme case where each image is its own domain, SSL might be suited to perform representation learning on such multi-domain data.


**Minor points**

- In Table 1, $\tilde{U}_{D+1}$ is undefined. I presume it is validation or test data from $D+1$ domain.
- In Section 3.1, using a linear shift in feature map of ResNets for mapping domains (to a "canonical domain"?) is not properly motivated. I assume this strategy has been used in prior works in language but it would be useful to motivate it in the context of domain adaptation problem.
- In Section 3.2, some of the symbols (eg. $\tau$, $q$) are introduced without proper definition. While I was able to understand them by going back and forth between current paper and referred paper, it would be helpful if the authors clarify the notations.
- Section 2.2 is a relatively minor detail given that the datasets used in the paper are not highly imbalanced and both observed accuracy and uniform accuracy are highly correlated to each other. Authors can consider moving this detail to the experiments section.



**Summary Of The Paper:**

The paper addresses the problem of learning a classification model on data from multiple domains, when explicit domain assignment for each data point is not provided.

To solve this problem, the paper proposes the data to be coming from 1 of K unknown (or latent) domains. A single 1x1 convolutional layer per latent domain is applied to feature maps in each residual layer of ResNet. A gating function (based on sparsemax) is proposed to decide which of the K convolutional layers will be applied. The entire network is learned jointly by minimizing the classification objective function.

**Summary Of The Review:**

Overall my opinion is that the paper explores an interesting problem of learning latent domains present in data, when no explicit domain information is provided. I feel the experiments performed in the paper can be improved a bit. I've scored the paper accordingly.

---

> ### Author Response · Authors · 2021-11-20
> **Re: Official Review of Paper228 by Reviewer LPa6**
>
> Thank you for your detailed review. We have addressed your concerns, see point by point below:
>
> * *“I specifically don't understand Why is the ResNet backbone kept frozen, and not fine-tuned along with rest of the parameters?”*
>
> We use this to provide a fair comparison to previous work in the multi-domain literature, e.g. Rebuffi et al., (2017/18) or Guo et al. (2019). These works showed that representations learned on large semantic tasks (e.g. ImageNet) are highly transferable and require minor adaptation strategies (such as small capacity adapters we use here), with the added benefit of being less prone to overfitting. Note our manuscript includes an ablation for finetuning all parameters (Appendix C, initial paragraph of page 16), which works less well in the latent domain scenario.
>
> * *“Why do the authors use same parameters across different methods/models?”*
>
> Most methods we compare against include highly non-trivial architectural configurations/hyperparameters, such as ResNeXt specifications in MLFN (Chang et al., 2018), or layer-specific latent clustering in MMLD (Matsuura & Harada, 2020). Because of this, we followed the recommendations in each respective publication and, where this was applicable (e.g. RA), used the exact same hyperparameters as in SLA. This prevents any unfavorable treatment of the baselines we compare against.
>
> * *“It is a bit surprising to me that all the proposed multi-domain approaches match perform worse when compared to latent-domain? I'd imagine that a model having a domain information for each datapoint would set an upper bound on the performance when compared to methods with no domain information. Is the presence of domain information hurting for the model performance?”*
>
> On first look this is a surprising result, this is true. Note however that multi-domain methods parametrize individual functions $f_d$ (e.g. adaptations in RA) for each domain, see also eq. (3). There cannot be any parameter-sharing between them in MD, since domain-specific parameters are by definition only optimized with data from an associated domain. This explains why performance is reduced for MD models, as some domains would have benefited from positive transfer with others (due to visual similarities). By virtue of its main assumption (domain label availability), MD cannot exploit such similarities, and this is an important criticism we make in our manuscript that motivates the LD setting.
>
> * *“The standard errors are not reported”*
>
> Variances of results were low across experiments/benchmarks. We have added an additional analysis of this aspect in Appendix E.
>
> * *“I would like to hear authors' take on connection between self-supervised learning approaches, and given latent domain learning. The SSL methods learn image representations using self-similarity. In an extreme case where each image is its own domain, SSL might be suited to perform representation learning on such multi-domain data.”*
>
> We believe this question connects to a lack of a canonical definition of what a domain is, see also page 1 (fourth paragraph). Studying such cases, e.g. via SSL, is extremely interesting, and we hope that such work can help the community develop a better sense of what a domain is and isn’t. Please note however that connecting SSL with multiple domains is a standalone research problem, and we therefore believe this deserves to be examined in its own future work.
>
> * *“Section 2.2 is a relatively minor detail given that the datasets used in the paper are not highly imbalanced and both observed accuracy and uniform accuracy are highly correlated to each other. Authors can consider moving this detail to the experiments section.”*
>
> The motivation for including the discussion around UAcc vs. OAcc was to prevent confusion for the reader at later stages. UAcc and OAcc represent two natural but different ways of thinking about performance in a latent domain setting, and we felt the need to clarify this aspect a priori.
>
> ---
>
> Thank you for your detailed comments and suggestions. We have updated notation that was perceived as confusing (e.g. for $\tilde U, \tau, q$), and edited Section 3.1 to clarify the aspect of affine transformations of pretrained internal representations.

---

> > ### Comment · Reviewer_LPa6 · 2021-11-29
> > **Response to the authors**
> >
> > Thank you for your detailed response. Having gone through other reviews and authors' responses, I retain my current score.

---

### Official Review · Reviewer_cM2p · 2021-11-02

**Correctness:** 3
**Technical Novelty And Significance:** 3
**Empirical Novelty And Significance:** 3
**Recommendation:** 6
**Confidence:** 3

**Main Review:**

Strength:
1. The proposed latent domain learning makes sense and technically sound to me. It also has great potential in real world applications.
2. Extensive experiments and sufficient analysis validated the approach empirically.
3. Writing is good and easy to follow.

Weakness:
1. It seems that the optimal value of hyperparameter K is different for different datasets. Is there any thorough methodology to pick a good value instead of using K from 2 to 5 or just 2 as experimented in the current draft?
2. For the results, it was averaged over 5 random initialization, what is the variance for each experiment? It is not sufficient enough to use the mean itself for comparison as there might be a very large variance that indicates the model is not robust.


**Summary Of The Paper:**

In this paper, the authors proposed latent domain learning for adaptation. Experiments on multiple benchmark datasets show improved result. Several visualizations also illustrate the effectiveness of the proposed approach.

**Summary Of The Review:**

Based on the above analysis, I am leaning towards acceptance for now.

---

> ### Author Response · Authors · 2021-11-20
> **Re: Official Review of Paper228 by Reviewer cM2p**
>
> Thank you for your positive review. We have addressed your main concerns in our revision:
>
> * *“It seems that the optimal value of hyperparameter K is different for different datasets.”*
>
> We have extended our analysis of the influence of K, and resulting variances from it, in our revision, see Appendix E.  Generally higher K tend to perform better. They saturate at some point, as all latent domain are eventually covered.
>
> * *“What is the variance for each experiment?”*
>
> As the results in Appendix E confirm, the variation of results is very low across the board. Moreover, as we state in our paper (Section 4), SLA can be optimized with standard methods, and is just as easy to optimize in practice as a standard ResNet.

---

> > ### Comment · Reviewer_cM2p · 2021-11-29
> > **Responses to Authors' Responses**
> >
> > Overall, the authors have addressed my main concerns and I am happy to keep my original rate of '6: marginally above the acceptance threshold'.

---

### Official Review · Reviewer_dZE1 · 2021-11-03

**Correctness:** 3
**Technical Novelty And Significance:** 2
**Empirical Novelty And Significance:** 2
**Recommendation:** 6
**Confidence:** 3

**Main Review:**

The paper is well written, easy to follow and the motivations and contributions are clear.  Multiple datasets are used for evaluation against multiple SOTA approaches.

SLA is referenced before defining it in 3.1

I am confused on how g_k is learned in Eq 4.  How does it learn domain related gating without domain labels?  Figure 5 shows that in PACS some domains seem to be learned (eg sketch) while others may not have been learned.  Could this suggest that sketch images required additional parameterization to increase performance due to task difficulty? It is mentioned that the goal is not to recover domain labels, but simply adding additional parameterization may be similarly beneficial in single domain data.  Have the authors tested whether the performance improvement extends to single domain settings?  The results are interesting and I feel the paper could benefit from a deeper analysis on what is being learned by the "domain" gates.  If they are not domain specific is this really multi-domain learning?. Also, how are the number of gates (K) chosen?

I would prefer a bit more examination on the results and what is being learned as well as comparisons to only learning the gating parameters and training the whole model together with the gates.

**Summary Of The Paper:**

The authors present the latent domain learning aproach to multi-domain learning without domain annotations by introducing sparse latent adaptation (SLA) for learning sparse domain gates without domain labels.

**Summary Of The Review:**

The paper is well motivated, clear and well evaluated with strong results.  It could benefit from a deeper analysis on what is being learned, how that contributes to different data settings and if novel aspects of datasets can be learned from this approach (eg does the model identify previously missed domains).  Overall a good paper that could be better with more detail.

---

> ### Author Response · Authors · 2021-11-20
> **Re: Official Review of Paper228 by Reviewer dZE1**
>
> Thank you for your positive feedback. We have addressed your concerns, and comment on each point below:
>
> * *“SLA is referenced before defining it in 3.1”*
>
> Thank you for noticing this, this oversight has been corrected in our revision.
>
> * *“I am confused on how g_k is learned in Eq 4. How does it learn domain related gating without domain labels?”*
>
> SLA and gates enable networks to account for hidden domains in data. This extends previous insights from multi-domain learning [Rebuffi et al., 2017/18] and the universality theorem [Bilen & Vedaldi, 2017], but bypasses the main assumption (domain labels required) in these works, which severely limit their applicability to real-world problems. Gates are learned end-to-end. During inference, if two image representations $x,y$ are very dissimilar (which could imply they are from different domains), then their gating decisions $g(x), g(y)$ would also become dissimilar in our model, so that they are processed by separate adaptations. Note our work uses a deep/layerwise module for this, different from previous one-stage clustering approaches [Hoffman et al., 2012; Xu et al., 2014].
>
> * *“Figure 5 shows that in PACS some domains seem to be learned (eg sketch) while others may not have been learned. Could this suggest that sketch images required additional parameterization to increase performance due to task difficulty?”*
>
> Both difficulty of each task and synergies between them contribute to the gating. However, in SLA the decision which domains to process separately (e.g. PACS sketch) is fully automated.
>
> * *“It is mentioned that the goal is not to recover domain labels, but simply adding additional parameterization may be similarly beneficial in single domain data. Have the authors tested whether the performance improvement extends to single domain settings?”*
>
> We evaluated this setting, as both the fairness benchmark (Appendix F) and the imbalanced CIFAR-10/100 results (Appendix G) are considered single domain data. SLA improves performance for single domains, because real-world single domain benchmarks are multimodal, having multiple (latent) domains (in particular semantic ones, e.g. animals vs. man-made objects in CIFAR benchmarks). Please note we updated our discussion of this aspect to better answer your question in our revision, see for example page 6 (second paragraph).
>
> * *“How are the number of gates (K) chosen?”*
>
> This can be tuned similar to other network components like depth/width. We discuss this aspect in new variance results in Appendix E of the revision, where the effect of K is displayed visually. Generally, higher K tend to perform better (c.f. Table 2 and 3), but this effect does saturate as eventually all important latent domains are covered (similar to e.g. ResNet depth).
>
> * *“It could benefit from a deeper analysis on what is being learned”*
>
> Please note we include an extensive analysis in this work. Amongst others, this includes an investigation of which layers exhibit sharing (Figure 3), or how sparsity is utilized in visual representations (Figure 4), alongside experiments on fairness benchmarks (Appendix F) and long-tailed recognition (Appendix G).
>
> * *“Is this really multi-domain learning?”*
>
> We believe multi-domain learning can and should be understood more broadly. Any complex and semantically meaningful learning problem has hidden domains (even single domain data, see above point). We hope our work will bring attention towards this aspect, and bring about the development of new latent domain methods in future research.

---

### Author Response · Authors · 2021-11-29
**General Author Response**

We thank the reviewers for their constructive feedback and are pleased that our manuscript has been well received overall:

**R1**: *"The paper is well motivated, clear and well evaluated with strong results."*

**R2**: *"The proposed latent domain learning makes sense and technically sound to me. It also has great potential in real world applications."*

**R3**: *"An important problem since labeling domains can be challenging and/or expensive in real-world applications."*

The main requests were for:

1. an additional analysis of the variation of results.
2. Clarifications regarding MD performance (in particular RA) in Table 2/3.

These points have been addressed in our revision, and we kindly ask reviewers to reconsider their final score in light of this.

---

### Decision · Program_Chairs · 2022-01-20

**Decision:**

Accept (Poster)

**Comment:**

The problem setting studied in this paper is an extension of the problem setting of multi-domain learning, where domain information is missing in training. This is an interesting and practical problem setting. However, regarding technical novelty, the contributions are relatively limited. Specifically, first, the overall idea is an extension of an existing multi-domain learning method [Rebuffi et al. 2017] by replacing the domain indicators with learnable gates. Second, the idea of introducing learnable gates is borrowed from some previous works. Third, the sparse activation is also based on an existing work of sparsemax. Though the authors claim sparsemax or sparse activation was only used in the NLP domain, this does not increase the technical novelty by applying sparsemax to the CV domain.

To be fair, the combination of the aforementioned techniques to solve the so-called latent domain learning problem looks reasonable, while the technical contribution is not significant. Overall, I feel slightly positive about this work and recommend a weak acceptance (it could be considered for publication only if there is space).